# dVLA: Diffusion Vision-Language-Action Model with Multimodal Chain-of-Thought

## Abstract

Vision-language-action (VLA) models have emerged as the next-generation framework in robotics, integrating visual perception, language reasoning, and robotic control into unified systems. In this paper, we present **dVLA**, a diffusion vision-language-action model with multimodal chain-of-thought. The dVLA optimizes visual reasoning, language comprehension, and robotic actions simultaneously through a unified diffusion-based objective. By harmonizing these modalities into a single cohesive framework, dVLA facilitates more effective cross-modal reasoning, enabling the model to generalize to novel instructions and objects. To ensure practical viability, we also integrate model acceleration methods that substantially decrease robot response times. Extensive evaluations in both simulation and the real world confirm that dVLA significantly outperforms current discrete and continuous VLA models, highlighting the potential of diffusion language model (DLM) based frameworks for robotics.

## 1 Introduction

Vision-language-action (VLA) models have emerged as the next-generation framework in robotics, integrating visual perception, language reasoning, and robotic control into unified systems (Black et al., 2024; Brohan et al., 2023; Kim et al., 2024; Hu et al., 2023; Liu et al., 2025a; Intelligence et al., 2025; Kim et al., 2025; Team et al., 2025; Bjorck et al., 2025; Zhao et al., 2025a;b; Zhen et al., 2024; Wen et al., 2025a;b; Zhou et al., 2025b; Wen et al., 2024). The development of VLA models has undergone two stages of evolution. In the first stage, a pre-trained vision–language backbone is used purely as a feature extractor, and the extracted features are mapped directly to robot actions. As vanilla VLA architectures proved inadequate for open-world instruction following and long-horizon tasks, a second-stage training paradigm co-trains on image–text data alongside action trajectories to preserve knowledge from the pre-trained VLM and, when necessary, to predict both sub-step reasoning and robot actions (Zhou et al., 2025b;a; Intelligence et al.; Driess et al., 2025). The sub-step reasoning, often referred to as Chain-of-Thought, grounds high-level instructions into low-level sub-steps, thereby offering improved guidance for action prediction. Recent works have also incorporated image generation capabilities into VLAs, enabling the prediction of subsequent images before generating actions, which is a visual form of Chain-of-Thought Zhao et al. (2025a); Cen et al. (2025). Leveraging images as intermediate reasoning steps offers a more detailed description of the next movement. Such approaches have demonstrated remarkable capabilities, enabling models to generalize to novel environments, adapt to new objects, and even complete tasks requiring complex reasoning, such as mathematical puzzle games (Zhou et al., 2025a; Zhao et al., 2025a).

Despite their promise, these models face several limitations. First, co-training visual-text data alongside robotic action data, each with distinct objectives, often results in gradient conflicts. Specifically, the gradients that enhance knowledge preservation and scene understanding may interfere with the model's ability to effectively learn robot actions, even when a separate module is dedicated to this task. Second, integrating image generation into auto-regressive Vision-Language Models (VLMs) is challenging due to the fundamental gap between training objectives and model architectures, which makes harmonizing multi-modal generation and understanding difficult. Consequently, VLAs struggle to fully exploit knowledge across all modalities, limiting their ability to capture the underlying physical laws that connect actions and generated images, even when equipped with an explicit Chain-of-Thought.

To address these challenges, we propose dVLA, a framework that jointly optimizes visual reasoning, image generation, and robotic manipulation under a unified diffusion-based objective. dVLA builds on MMaDA (Yang et al., 2025), an advanced model in discrete diffusion language models that unifies multimodal understanding and generation through a consistent discretization strategy, employing modality-specific tokenizers. To extend this foundation to actions, we adopt FAST (Pertsch et al., 2025) to encode action sequences into compact discrete tokens, enabling dVLA to leverage pretrained visual–textual knowledge for generating executable actions. However, simply discretizing actions and applying a unified training objective is insufficient. Such an approach exploits only MMaDA's multimodal understanding capabilities while neglecting its core strength—multimodal generation. To overcome this limitation, we introduce a multimodal Chain-of-Thought (CoT) training paradigm, in which dVLA is required to simultaneously generate subgoal images (visual CoT), textual reasoning, and action sequences. Concretely, during training we randomly mask tokens not only from actions but also from subgoal images and textual reasoning, and the model is required to reconstruct them across all available modalities. This design encourages dVLA to learn a shared parameter space, ensuring strong consistency between predicted subgoal images and actual execution outcomes. Empirically, we observe that dVLA can even forecast failed execution images that precisely match real-world failures, suggesting that it learns not just to generate fixed sub-goal images but also to capture the underlying physical laws governing action and perception.

In this paper, we conduct a comprehensive evaluation of dVLA through rigorous experimental analysis. On the LIBERO benchmark, dVLA achieves an average success rate of 96.4%, consistently outperforms both discrete and continuous action policies, and achieves state-of-the-art performance. We further validate our approach on a real Franka robot across a wide range of tasks, including the challenging bin-picking task, which requires multi-step planning to complete. The results demonstrate dVLA's superior ability to handle the complexities of real-world scenarios, highlighting its potential to significantly advance the capabilities of vision-language-action robotic systems. Since multimodal CoT prediction increases inference cost, we introduce two acceleration strategies: prefix attention mask and KV caching. These optimizations yield up to $\sim 2\times$ speedup in both real-world tasks and the LIBERO benchmark, with only marginal performance degradation.

## 2 RELATED WORK

**Diffusion Language Models.** Modern state-of-the-art Vision-Language Models (VLMs) are predominantly built upon autoregressive large language models (LLMs), which rely on an autoregressive training objective (Achiam et al., 2023; Liu et al., 2024c;b; Wang et al., 2024; Team, 2024). Recent advances in discrete diffusion language models (DLMs) have demonstrated their potential as superior alternatives for language modeling (Austin et al., 2021; Sahoo et al., 2024; Lou et al., 2023; Nie et al., 2025). These models achieve performance comparable to autoregressive models while offering distinct advantages, such as flexible speed-quality trade-offs and enhanced controllability. Furthermore, recent studies have begun exploring the integration of discrete diffusion language models (DLMs) with visual question answering (VQA) capabilities. For instance, LaViDa (Li et al., 2025) adopts a standard LLaVA-like architecture with a two-stage training framework to achieve this. MMaDA (Yang et al., 2025) introduces a unified diffusion-based foundation model that combines textual reasoning, multimodal understanding, and generation within a single probabilistic framework. Some other works have explored the DLMs to robotics that Discrete Diffusion VLA (Liang et al., 2025) adopts the discrete diffusion training strategy to an off-the-shelf VLA, while LLaDA-VLA (Wen et al., 2025c) directly trains a DLM to predict action tokens. In this work, we investigate the potential of DLMs for robot manipulation and multi-modal Chain-of-Thought, aiming to leverage their unique properties for more robust and interpretable policy learning.

**Vision-Language-Action Model.** Vision-Language-Action (VLA) models build on pre-trained Vision-Language Models (VLMs) together with specialized action experts/heads to generate robot actions, and have become a prominent approach for exploiting vast heterogeneous data for scalable policy learning (Bommasani et al., 2021; Black et al., 2024; Team et al., 2024; Chi et al., 2023). Despite state-of-the-art results across diverse tasks and embodiments, most VLAs learn a direct mapping from observations to actions without explicit intermediate reasoning, which limits generalization to open-world scenarios and long-horizon tasks (Black et al., 2024; Wen et al., 2025b; Bjorck et al., 2025). Recent work leverages the auto-regressive reasoning capabilities of language models to decompose long-horizon tasks into stepwise subgoals and then condition action generation on

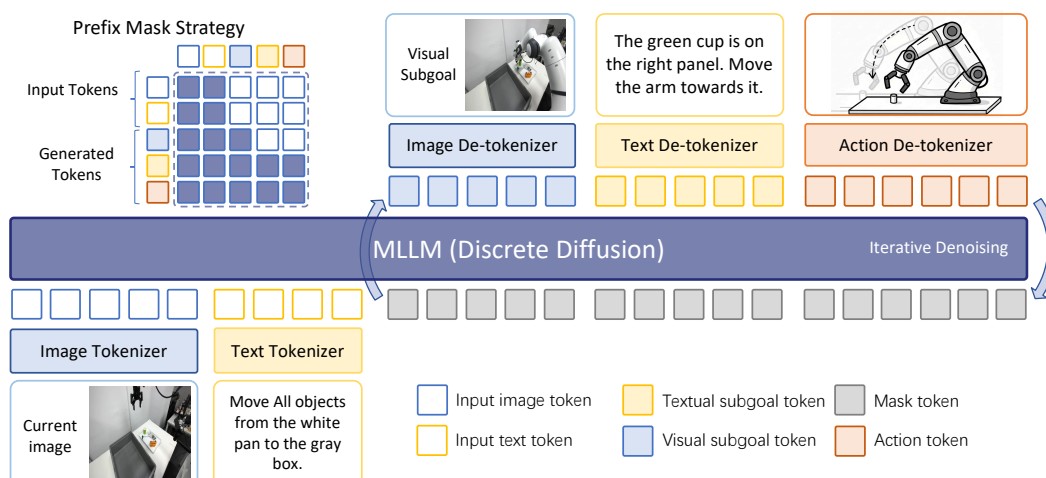

Figure 1: The architecture of dVLA. We adopt a discrete diffusion language model as a backbone and separate tokenizers for each modality.

these plans (Wen et al., 2024; 2025a; Intelligence et al.; Driess et al., 2025; Liu et al., 2025a). However, the reasoning and control components are typically optimized separately, leading to plan–act misalignment between task-level reasoning and execution-level control. We propose dVLA, which unifies reasoning and action under a single diffusion-based training objective, enabling joint optimization and tighter coupling between planning and control, thereby yielding more coherent and generalizable policies.

**Multi-modal Chain-of-Thought Reasoning.** Step-by-step reasoning has emerged as a critical capability enabling large language models (LLMs) to tackle complex tasks effectively. Prompting LLMs to "think step-by-step" about the problem before formulating an answer can significantly improve their performance (Lu et al., 2023; Yang et al., 2025). This chain-of-thought (CoT) paradigm has become a standard technique in language modeling and vision-language model training (Chung et al., 2024; Zhou et al., 2024). Recent work has extended textual reasoning to robotic control domains (Intelligence et al.; Wen et al., 2025a). However, existing approaches typically employ two distinct training objectives: (1) a discretized token prediction objective for reasoning and (2) a continuous action prediction objective for robotic control. This decoupled optimization creates a fundamental optimization gap that hinders effective cross-modal learning and limits the potential synergy between high-level reasoning and low-level control (Driess et al., 2025). On the other hand, the action-prediction objective requires the model to predict the intermediate noise, while the next-token prediction objective requires it to estimate the next-token distribution. The two objectives are naturally disparate. In this paper, we resolve this issue by casting vision, language, and action prediction as a single diffusion-based denoising objective, thereby harmonizing cross-modal generation and further improving action prediction through a shared latent-space CoT.

## 3 METHOD

In this section, we present dVLA, designed for multimodal chain-of-thought (CoT) generation and action prediction. We first introduce the unified training objective 3.1, followed by a detailed description of the architecture 3.2. Next, we define multimodal CoT and outline the approach to achieving it 3.3. Finally, we introduce two acceleration strategies for real-time inference 3.4.

### 3.1 UNIFIED PROBABILISTIC FORMULATION FOR TRAINING

dVLA aims to tackle the challenge of learning a unified model capable of simultaneously generating multimodal chain-of-thought reasoning (including subgoal image generation and reasoning) and action prediction. In contrast to current methods that rely on separate foundation models for each component, we adopt a unified approach by aligning the training objectives through a consistent discrete strategy and discrete diffusion modeling.

**Unified discrete strategy.** dVLA processes data from three distinct modalities: vision, text, and action. Building upon MMaDA (Yang et al., 2025), dVLA employs the same tokenization approach, encoding raw images and text into discrete tokens using MAGVIT-v2 (Yu et al., 2023) for vision and the LLaDA text tokenizer (Nie et al., 2025) for textual data. For action tokenization, we utilize the Fast tokenizer (Pertsch et al., 2025), which discretizes continuous actions using Discrete Cosine Transform (DCT) (Ahmed et al., 2006) and compresses tokens with Byte Pair Encoding (BPE) (Gage, 1994).

**Discrete diffusion modeling.** After consistent discrete tokenization, the input sequences can be represented as $x = \{o, l, s, o_{\text{goal}}, r, a_{\text{chunk}}\}$, where $o$ denotes the current observations, $l$ represents the language instructions, $s$ refers to the current robot state, $o_{\text{goal}}$ indicates the visual reasoning (subgoal image) corresponding to a few frames ahead of the current time, $r$ refers to the current textual reasoning, and $a_{\text{chunk}}$ represents the action chunk to be executed. During training, tokens from different modalities are randomly masked with a certain probability, then dVLA must predict all masked tokens based on other unmasked tokens. Formally, the training objective for dVLA is defined as:

$$\mathcal{L}_{\text{unify}}(\theta) = -\mathbb{E}_{t, x_0, x_t} \left[ \frac{1}{t} \sum_{i=1}^{L} \mathbf{I}[x_t^i = \texttt{[MASK]}] \log p_\theta(x_0^i | x_t) \right], \tag{1}$$

where $x_0$ is ground truth, the timestep $t$ is sampled uniformly from $[0, 1]$, $L$ denotes the sequence length of $x$ and $x_t$ is obtained by applying the forward diffusion process to $x_0$. $\mathbf{I}[\cdot]$ denotes the indicator function to ensure that the loss is computed only over the masked tokens.

## 3.2 THE dVLA ARCHITECTURE

The overall architecture of dVLA is shown in Figure 1. dVLA is initialized from MMaDA (Yang et al., 2025), a unified diffusion model for image generation and multimodal understanding (Xie et al., 2024). At its core is a discrete diffusion modeling objective that predicts both visual and textual tokens using the same diffusion decoding process. Specifically, dVLA first employs different tokenizers for each modality. For image tokenization, MAGViT-v2 (Yu et al., 2023) converts raw image pixels into discrete semantic tokens, with a compression ratio of 16 and a codebook size of 8192. Given input images of size $256 \times 256$ and $512 \times 512$, MAGViT-v2 generates 256 and 1024 tokens, respectively. For text tokenization, LLaDA's tokenizer (Nie et al., 2025) maps raw language to discrete tokens in a vocabulary of size 126,464. For action tokenization, the Fast tokenizer (Pertsch et al., 2025) encodes actions into discrete tokens with a vocabulary size of 2048. To accommodate all tokens from different modalities, the original vocabulary size is expanded from 126,464 to 136,704. All texts, actions, and images are discretized into tokens and trained under the same discrete diffusion modeling.

## 3.3 MULTI-MODAL CHAIN-OF-THOUGHT (CoT) REASONING

dVLA's unified tokenization allows it to jointly model vision, language, and actions through a multi-modal CoT mechanism. This step-by-step reasoning is critical for translating high-level instructions into executable actions.

**Multi-modal CoT Data.** The input sequence combines $M$ images, language instructions, and robot state, followed by multi-modal CoT tokens (sub-goal image and reasoning text), and finally action tokens:

$$\overbrace{\texttt{[BOS]}\underbrace{\texttt{[BOI]\{image\}[EOI]}}_{\times M}\texttt{\{text\}\{state\}}}^{Observation\ and\ instrcution}\overbrace{\texttt{[BOI]\{Subgoal\}[EOI]\{Reasoning\}}}^{Multi-modal\ CoT\ Reasoning}$$

$$\underbrace{\texttt{[BOA]\{action\}...\{action\}[EOA]}}_{\mathcal{L}_{action}}\texttt{[EOS]}$$

where $\{\texttt{text}\}$ is the overall language instruction (e.g., "Move any object from the panel to the box.").The robot state is discretized and input to the model as text tokens, same as Driess et al.

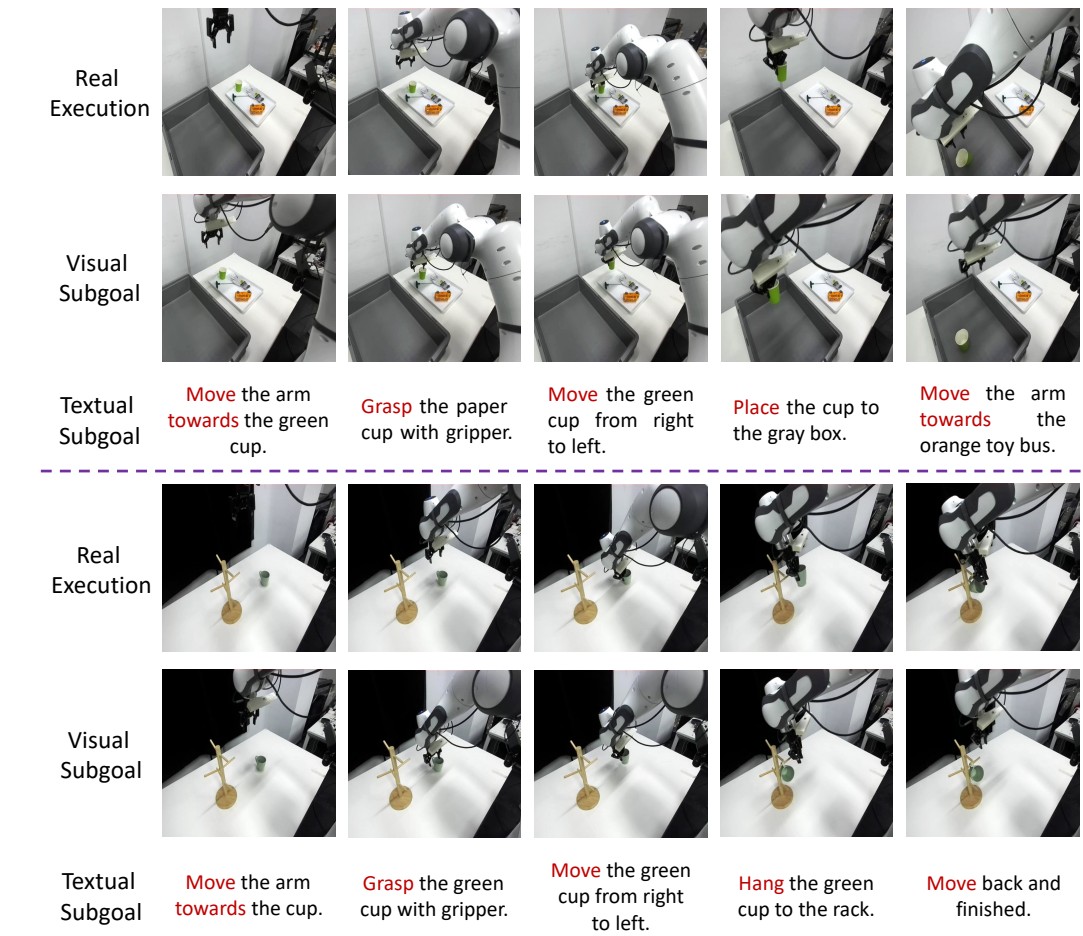

Figure 2: Examples of multimodal Chain-of-Thought on real robot tasks.

(2025). Given these inputs, dVLA should first reason about a subgoal image of the future state. Next, the model predicts a high-level subtask to instruct the model for what to do (e.g., "pick up the toy blue car"). Finally, it generates discretized action tokens. This multi-modal CoT enables the model to (1) perform visual reasoning via subgoal prediction and (2) decompose the task into interpretable subtasks before generating low-level actions. Figure 2 presents illustrative examples of our multimodal Chain-of-Thought (CoT) process. During inference, dVLA generates two parallel outputs: (i) a visual CoT that depicts the intended physical movement in detail, and (ii) a textual CoT that provides fine-grained, step-by-step instructions. Subsequently, dVLA grounds these multimodal reasoning steps to produce a concrete and executable action.

## 3.4 ACCELERATION STRATEGIES

To enhance the inference efficiency of dVLA, we adopt two acceleration strategies: a prefix attention mask and a KV caching method. The prefix attention mask is incorporated during training to better preserve model performance, while the KV caching approach is a plug-and-play technique applicable at inference. Combined, these strategies deliver substantial speedups, achieving $2\times$ on both the LIBERO benchmark (Liu et al., 2024a) and our real-world bin-picking task.

**Prefix Attention Mask.** As described in Section 3.1, we build dVLA upon MMaDA (Yang et al., 2025), which typically exhibits slower inference than autoregressive models because it cannot leverage KV caching (Nie et al., 2025). Following the approach in LaViDa (Li et al., 2025), we adopt a prefix attention mask for partial KV caching. Specifically, our architecture utilizes a blockwise causal attention mask with two blocks: $[o, l, s]$ and $[o_{\text{goal}}, r, a_{\text{chunk}}]$. We apply full bidirectional attention within each block, with tokens in one block restricted from attending to tokens in subsequent

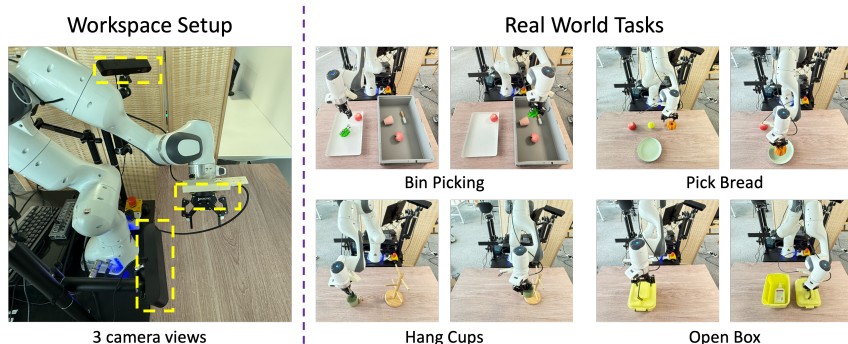

Figure 3: The experiment setup and real-world task suite.

blocks. The first block contains multi-view images and instructions, which are all input tokens. The second block includes discretized subgoal image tokens, reasoning tokens, and action tokens, allowing action tokens to attend to other modalities.

**KV Caching.** To further accelerate diffusion-based denoising, we incorporate the training-free KV Caching technique from dLLM-Cache (Liu et al., 2025b). This method leverages the observation that, across denoising steps, changes in key-value features and attention outputs are minimal. Instead of recomputing them at every step, dLLM-Cache caches intermediate results and refreshes them at a lower frequency. This reduces computational overhead while maintaining high accuracy, enabling dVLA to operate efficiently in real-time robotic settings.

### 3.5 EXPERIMENTAL SETUP

**Robot Setup.** We perform evaluation on both simulation and real-world tasks (shown in Fig 3). For simulation, we use the LIBERO benchmark (Liu et al., 2024a) to evaluate all policies for learning lifelong in robot manipulation. Additionally, we evaluate all policies on 4 tasks with a 7-DoF Franka robot arm as show in 3. We used two external ZED cameras and a Realsense 435i wrist camera to obtain real-world visual information.

**Baselines.** We compare our dVLA to state-of-the-art models, including continuous action policies and discretized action policies. Continuous action policies generate action chunks by progressively denoising a Gaussian noise action chunk into an executable action chunk (Ho et al., 2020; Lipman et al., 2022). As baselines for this type of policy, we select Diffusion Policy (Chi et al., 2023), GR00T-N1 (Bjorck et al., 2025) Octo (Octo Model Team et al., 2024), DiT Policy (Hou et al., 2024), and $\pi_0$ (Black et al., 2024). In contrast, discrete action policies mainly tokenize continuous actions into a discrete form to align with current auto-regressive language models or diffusion language models. These methods predict discretized action tokens using the next-token prediction or parallel decoding, which are then denormalized to continuous actions. We select OpenVLA (Kim et al.), CoTVLA Zhao et al. (2025a), OpenVLA-OFT Kim et al. (2025), WorldVLA (Cen et al., 2025), Discrete Diffusion VLA (Liang et al., 2025) as baselines. Additionally, we use vanilla dVLA as a baseline, which predicts only discretized action tokens for establishing the performance of multi-modal Chain-of-Thought (CoT).

**Training Datasets.** We evaluate on the LIBERO simulation benchmark (Liu et al., 2024a), which consists of four task suites: LIBERO-Spatial, LIBERO-Object, LIBERO-Goal, and LIBERO-Long. Each suite offers 10 diverse tasks, with 50 human-teleoperated demonstrations per task, challenging the robot's abilities in spatial reasoning, object manipulation, and goal fulfillment. We regenerate all demonstrations at an increased resolution of $256 \times 256$ pixels and then filter out the demonstrations that fail to complete the task following OpenVLA. For real-world tasks, we collect 1100 trajectories in total, including 4 different tasks as shown in 3. The details of each task are listed as follows:

- **Bin Picking.** We collect 600 trajectories. This is a long-horizon robotics task where the goal is to transfer all objects from the right tray to the gray box. In each trajectory, 3-5 randomly selected objects are individually placed into the box. This scenario presents a cluster grasping challenge, as the presence of multiple objects can interfere with the policy's ability to predict accurate grasps for individual items.

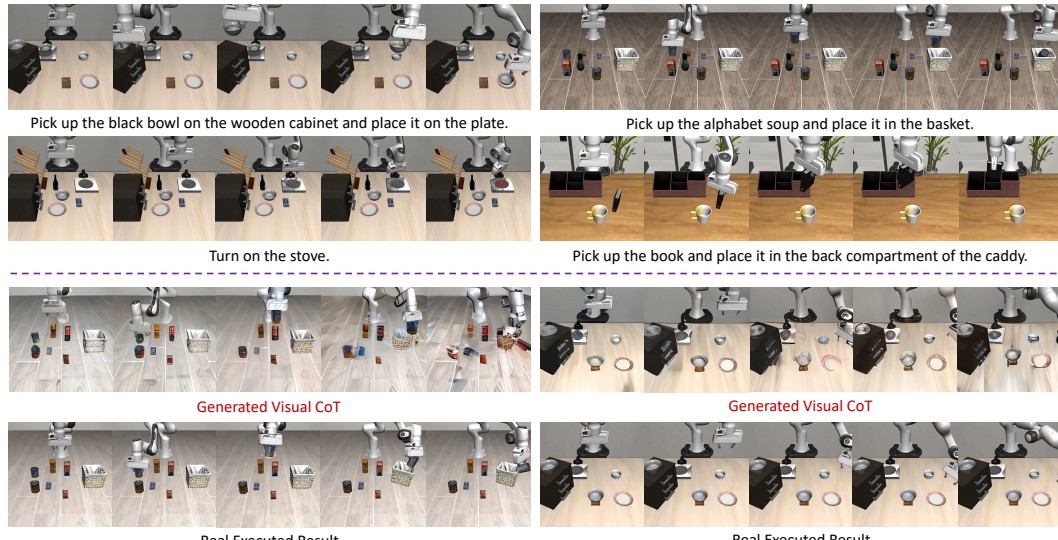

Figure 4: Qualitative results on LIBERO simulation. **Top**: The successful execution results. **Bottom**: Failure execution results and corresponding visual CoT.

- **Open Box.** There are 100 trajectories in total for this task. The robot must accurately grasp the handle and lift the lid clear of the box. Then place the lid in an empty place.

- **Hang Cups.** We leverage 200 trajectories. The robot must pick up a cup and hang it on a rack. This is a relatively challenging task because the cup's small handle requires very precise alignment for successful hanging.

- **Pick up the object and place to plate(Pick&place Object).** 200 trajectories. The robot must pick up a specific object and place it on the plate based on language instructions. This task presents a significant challenge, as the policy must learn to map language instructions to the correct object and to associate each object with its corresponding motion sequence.

**Training and Evaluation Details.** We finetune dVLA on both the LIBERO dataset and real-world data using the same training pipeline as MMaDA. All input images are resized to a resolution of $256 \times 256$ to reduce the input token sequence length. Our multimodal Chain-of-Thought (CoT) data consists of two components: visual subgoal reasoning and textual reasoning. For visual subgoal reasoning, dVLA predicts sub-goal images at future timestep $t$, uniformly sampled from the range $[0.9C, 1.1C]$, where $C$ denotes the action chunk length. We set $C = 5$ for LIBERO tasks and $C = 50$ for real-world tasks. To accelerate inference, we resize these sub-goal images to $256 \times 256$ and restrict dVLA to predicting only top-view camera images. We employ classifier-free guidance (scale = 3.5) to balance diversity and quality in the generated subgoal images. For textual reasoning, we use SEED-1.5VL (Guo et al., 2025) to generate video segmentation annotations at 3-second intervals, which are designed for long-horizon tasks such as bin picking. For simpler tasks, we omit language reasoning to further speed up inference.

## 4 EXPERIMENTS

This section evaluates dVLA's effectiveness for robot control across various manipulation tasks, addressing three key questions: (1) How does our framework compare with state-of-the-art baselines across different tasks? (2) Does multi-modal chain-of-thought reasoning improve dVLA's performance? (3) How do the acceleration strategies affect performance and inference speed?

### 4.1 MAIN EVALUATIONS RESULTS

**Experimental results on LIBERO.** As shown in table 1, we report success rate (SR) across four LIBERO task suites. The qualitative results are displayed in Figure 4. dVLA achieves the best average success rate of 96.4% and outperforms all continuous and discrete action policies. Specifically,

Table 1: **Experimental results on LIBERO benchmark.** We evaluated our model on 10 LIBERO tasks, with 50 trials per task, for a total of 500 trials. The success rate is the total number of successful trajectories out of 500.

| Methods / Tasks | MMCoT | | Libero-Spatial | Libero-Object | Libero-Goal | Libero-Long | Average |
|---|---|---|---|---|---|---|---|
| | Textual CoT | Visual CoT | SR(%) | SR(%) | SR(%) | SR(%) | SR(%) |
| **Continuous Action Policy** | | | | | | | |
| Diffusion Policy (Chi et al., 2023) | ✗ | ✗ | 78.3 | 92.5 | 68.3 | 50.5 | 72.4 |
| Octo (Octo Model Team et al., 2024) | ✗ | ✗ | 78.9 | 85.7 | 84.6 | 51.1 | 75.1 |
| DiT Policy Hou et al. (2024) | ✗ | ✗ | 84.2 | 96.3 | 85.4 | 63.8 | 82.4 |
| $\pi_0$ (Black et al., 2024) | ✗ | ✗ | 96.8 | **98.8** | 95.8 | 85.2 | 94.2 |
| GR00T-N1 (Bjorck et al., 2025) | ✗ | ✗ | 94.4 | 97.6 | 93.0 | 90.6 | 93.9 |
| OpenVLA-OFT (Continuous)(Kim et al., 2025) | ✗ | ✗ | 96.9 | 98.1 | 95.5 | 91.1 | 95.4 |
| **Discrete Action Policy** | | | | | | | |
| OpenVLA (Kim et al.) | ✗ | ✗ | 84.7 | 88.4 | 79.2 | 53.7 | 76.5 |
| OpenVLA-OFT (Discrete)(Kim et al., 2025) | ✗ | ✗ | 96.2 | 98.2 | 95.6 | 92.0 | 95.5 |
| CoTVLA (Zhao et al., 2025a) | ✗ | ✓ | 81.13 | 87.5 | 91.6 | 87.6 | 69.0 |
| WorldVLA ($512 \times 512$) (Cen et al., 2025) | ✗ | ✓ | 87.6 | 96.2 | 83.4 | 60.0 | 81.8 |
| Discrete Diffusion VLA (Liang et al., 2025) | ✗ | ✗ | 97.2 | 98.6 | 97.4 | 92.0 | 96.3 |
| Vallina dVLA | ✗ | ✗ | 90.2 | 93.1 | 92.8 | 83.0 | 89.8 |
| dVLA | ✓ | ✓ | **97.4** | 97.9 | **98.2** | **92.2** | **96.4** |

Table 2: **Experimental results for real-world tasks.** We evaluate each method on four real-world robotic tasks, ranging from a simple pick-and-place to a long-horizon bin picking scenario. Each method is tested for 10 trials per task (40 trials total), and we report the total number of successful trajectories.

| Methods / Tasks | MMCoT | | Bin Picking | Open Box | Hang Cups | Pick&place Object | Average |
|---|---|---|---|---|---|---|---|
| | Textual CoT | Visual CoT | SR(%) | SR(%) | SR(%) | SR(%) | SR(%) |
| **Continuous Action Policy** | | | | | | | |
| Diffusion Policy (Chi et al., 2023) | ✗ | ✗ | 2/10 | 4/10 | 4/10 | 4/10 | 14/40 |
| GR00T (Bjorck et al., 2025) | ✗ | ✗ | 4/10 | 5/10 | 4/10 | 5/10 | 19/40 |
| **Discrete Action Policy** | | | | | | | |
| OpenVLA (Kim et al.) | ✗ | ✗ | 2/10 | 3/10 | 5/10 | 4/10 | 14/40 |
| Vallina dVLA | ✗ | ✗ | 5/10 | 5/10 | 6/10 | 5/10 | 21/40 |
| dVLA | ✓ | ✓ | **7/10** | **5/10** | **7/10** | **7/10** | **26/40** |

dVLA achieves 97.4%, 97.9%, 98.2%, 92.2% in the spatial task suite, object task suite, goal task suite, and long task suite, respectively. For continuous action baselines, dVLA outperforms Open-VLA (Cont-Diffusion) by 1.0%, GR00T-N1 by 2.5% $\pi_0$ by 2.2%, respectively. For discrete action baselines, dVLA outperforms WorldVLA by 14.6%, CoTVLA by 27.4%, and Discrete Diffusion VLA by 0.1%, respectively. These results suggest that dVLA attains benefits from a unified training objective and model architecture.

**Experimental results on real-world tasks.** As detailed in Table 2, all methods were finetuned in a multi-task setting and evaluated over 10 trials per task. We categorized baselines into continuous and discrete action policies based on different action representations. While continuous baselines like GR00T achieved a decent 60% success rate in both the hang cups and pick-and-place tasks, our dVLA performed slightly better, reaching a 70% success rate. Diffusion Policy (DP) and OpenVLA recorded an average success rate of 35% and particularly struggled with the bin-picking task, where cluttered scenarios made precise grasping predictions more challenging. Ultimately, dVLA delivered the highest average success rate of 65%, consistently outperforming all continuous and discrete baselines.

## 4.2 MULTI-MODAL CHAIN-OF-THOUGHT REASONING IMPROVE DVLA'S PERFORMANCE

**Multi-modal CoT.** To assess the impact of multi-modal CoT reasoning, we evaluated the performance of vanilla dVLA (dVLA without explicit multi-modal CoT). As reported in Tables 2, vanilla dVLA still achieved a commendable 52.5% success rate, outperforming Diffusion Policy (DP) and OpenVLA, which underscores the inherent efficacy of our core dVLA approach. Furthermore, integrating multi-modal CoT reasoning improved dVLA's average success rate by 12.5%, reaching 65%. This gain further validates the effectiveness of our multi-modal CoT framework in enhanc-

Table 3: **Effect of KV Caching and Prefix Attention Mask.** We report the inference speed (actions per second) and task success rate (SR) between the full attention and our accelerated strategies on boththe LIBERO simulation and real-world bin picking tasks.

| Methods | LIBERO | | | Real World | | |
|---|---|---|---|---|---|---|
| | Spatial | Object | Actions / s (↑) | Bin Picking | Hang Cups | Actions / s (↑) |
| Full Attention | 97.4 | 97.9 | 1.3 Hz | 7/10 | 8/10 | 1.5 Hz |
| Prefix Attention + KV Caching | 96.9 | 97.3 | **2.9 Hz** | 7/10 | 7/10 | **3 Hz** |

ing robotic manipulation. Specifically, in the bin-picking task, empirical observations revealed that vanilla dVLA's grasping pose predictions suffered from multi-object interference, often attempting to grasp the space between objects—a deficiency even more pronounced in OpenVLA. Conversely, dVLA leveraged its unified understanding and generation capabilities from MMaDA to create sub-goal images that imagined an object will be grasped and provided language reasoning to indicate the target object. This explicit multi-modal Chain-of-Thought (CoT) enabled the policy to predict more precise grasping poses, significantly reducing inter-object interference. For the LIBERO simulation, we observed a salient improvement when utilizing multi-modal CoT. As reported in Table 1, dVLA reaches an averaged SR 96.4% against 89.8% for vallina dVLA with a 6.6 point gain. Overall, our dVLA achieved the best results, validating the effectiveness of multi-modal CoT reasoning for VLA tasks.

**dVLA can prevent unsafe actions via multimodal CoT.** During evaluation of our model on the LIBERO task suites, dVLA sometimes delivers unsafe actions and fails to complete the tasks. We empirically observed that the visual CoT generated by dVLA surprisingly aligns with the real execution results. Specifically, as shown in the bottom of Figure 4, the visual CoT on the left exhibits that the object is stuck between the gripper and the edge of the box, while the right one showcases that the robot moves in the wrong direction and struggles to move back. Both visual CoTs accurately predict the unsafe behaviors of real executed actions, indicating that dVLA can predict not only correct subgoal images but also the wrong execution results of unsafe actions. This is mainly due to the unified discrete diffusion training strategy that dVLA predicts masked tokens based on all available tokens across different modalities. Thus, dVLA naturally learns a unified and consistent representation that can better ground multimodal Chain-of-Thought into concrete actions.

**Effect of acceleration strategies.** DLMs (Diffusion Language Models) typically cannot utilize key-value (KV) cache during inference due to their bidirectional attention mechanism in training. Thus, we employ two acceleration strategies to improve the inference speed of dVLA. As shown in Table 3, we compare the inference speed and task success rate on both LIBERO and real-world scenarios. The results demonstrate that using prefix attention combined with KV caching significantly boosts inference speed from 1.5 Hz to 3 Hz, with only a marginal performance cost, highlighting the effectiveness of our acceleration strategies in enhancing dVLA's real-time performance.

## 5 CONCLUSION

In this work, we introduced dVLA (diffusion Vision-Language-Action Model), the first vision-language-action framework built on diffusion language models (DLMs). dVLA addresses the key challenge of learning a unified architecture that can jointly perform multimodal Chain-of-Thought reasoning—including subgoal image synthesis and textual reasoning—while simultaneously predicting actions. Moreover, dVLA demonstrates a strong grasp of the implicit physical laws underlying actions, as it can forecast future images that accurately reflect the real execution outcomes of unsafe actions. This highlights that a unified model framework with a shared training objective enables consistent reasoning and generation across modalities. To mitigate the inference overhead of multimodal CoT prediction, we further introduced two acceleration strategies: a block-wise causal attention mechanism for training and KV caching for inference. Together, these advances establish a solid foundation for applying unified DLMs in robotics and pave the way for future research in this direction.

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
