APPENDIX

## A  STATEMENT ON LLM USAGE

In accordance with ICLR's policy on the use of Large Language Models (LLMs) during paper preparation, we declare the extent and nature of LLM involvement in this work. We employed an LLM (ChatGPT, GPT-5 by OpenAI) solely for *text refinement* purposes, including improving grammar, enhancing wording clarity, and rephrasing sections to meet academic writing standards. No scientific content, experimental results, or novel ideas were generated by the LLM; all technical contributions—including model design, experiments, and analysis—were conceived, implemented, and verified by the authors.