# OpenReview forum: "dVLA: Diffusion Vision-Language-Action Model with Multimodal Chain-of-Thought"
_ICLR.cc/2026/Conference — Submitted to ICLR 2026_

### Official Review · Reviewer_pWAW · 2025-10-27

**Soundness:** 3
**Presentation:** 3
**Contribution:** 2
**Rating:** 4
**Confidence:** 4

**Summary:**

This paper introduces dVLA, a novel Vision-Language-Action (VLA) model built upon a discrete diffusion language model (DLM) backbone. The authors identify key limitations in existing VLA frameworks, namely the gradient conflicts arising from co-training with disparate objectives (e.g., for action prediction and vision-language understanding) and the architectural difficulty of integrating multimodal generation, such as visual Chain-of-Thought (CoT), into standard autoregressive models.  To address this, dVLA proposes a unified probabilistic framework where all modalities—including observations, language instructions, textual reasoning, visual subgoal images, and robot actions—are discretized into a shared token space and optimized jointly via a single diffusion-based denoising objective. The core contribution is a multimodal Chain-of-Thought training paradigm where the model is trained to reconstruct randomly masked tokens across all modalities, forcing it to learn consistent and physically grounded cross-modal representations.

**Strengths:**

- The core idea of unifying multimodal understanding and generation (including actions) under a single diffusion-based training objective is elegant.
- The authors proactively address the primary drawback of diffusion models—inference latency—by incorporating and evaluating acceleration strategies. This shows a thoughtful consideration of the practical requirements for deploying robotic policies and strengthens the overall contribution.
- The paper is easy to follow.

**Weaknesses:**

- The multimodal CoT paradigm requires a complex data pipeline that involves generating visual subgoals (future frames) and textual reasoning (using a model like SEED-1.5VL). The paper could provide more details on the practicalities and potential costs or biases associated with this data generation process, which is a critical component of the overall framework.
- The paper makes the strong claim that the model learns "underlying physical laws" because it can predict failure images. While a fascinating result, this could be an overstatement. The model may be learning to recognize and reconstruct common failure modes from the training data's statistics, rather than developing a true causal or physical understanding. This claim should be toned down or supported with more targeted evidence (e.g., generalization to novel failure types).
- While the paper shows an example of predicting failure, it does not provide a broader analysis of the model's own failure modes. What are the common failure patterns of dVLA itself during evaluation? Does it fail on reasoning, visual grounding, or action precision? This would provide valuable insights for future work.

**Questions:**

Please see Weakness.

---

> ### Author Response · Authors · 2025-12-03
> **Response**
>
> ### 1. Details on the practicalities and potential costs or biases associated with this data generation process.
> Thank you for pointing this out. Our data generation pipeline is as follows: visual subgoals are obtained by uniformly sampling future frames within the range [0.9C, 1.1C], where C denotes the action chunk length. For textual reasoning, we generate annotations using SEED-1.5VL at 2-second intervals. We ask the SEED-1.5VL to label the detailed substep annotations and some sommonsense reasonsings.
>
> The main source of bias is that the textual reasoning annotations in SEED-VL are not perfectly aligned with the actual transition points in the executed trajectories. Due to timestamp noise during video annotation, it is difficult to precisely locate the exact moment when a subtask changes, so the textual annotations may slightly lag behind or precede the true robot state. The annotation cost of SEED-1.5VL is also very low: at approximately 0.09 RMB per 60-second video, annotating 2,000 episodes with an average length of 30 seconds would cost only about 90 RMB in total.
>
> ### 2. Can dVLA generalize to novel failure cases?
> Thank you for your insightful comment. As mentioned on page 6, line 317, our training dataset filters out failed trajectories and focuses only on successful demonstrations. Thus, when dVLA predicts failure images, it is not replaying common failures from memory, but rather extrapolating from successful trajectories using the learned relationship between visual states and action outcomes. We agree that the phrase 'underlying physical laws' is too strong. Our qualitative results instead suggest that dVLA learns a consistent alignment between vision and action modalities and can generalize this alignment to novel failure modes. We will tone down this claim in the paper accordingly.
>
> ### 3. Does dVLA fails on reasoning, visual grounding, or action precision?
>
> Thank you for raising this point. In our current experiments, the most prominent failure mode of dVLA is not due to high-level reasoning or visual grounding, but to issues in action decoding. Because the FAST-encoded action tokens have variable length, even though we feed a fixed-size action chunk during training, the model occasionally fails to generate the correct end-of-action token '\<eoa\>' at inference time. This can result in action sequences with unexpected lengths or wrong token patterns, which in turn cause decoding errors or strange low-level motions that prevent the robot from completing the task, despite the visual and textual CoT being reasonable. A more systematic analysis and mitigation of these action-decoding failures (e.g., with improved tokenization or decoding constraints) is an important direction for future work.

---

### Official Review · Reviewer_eZGi · 2025-10-28

**Soundness:** 2
**Presentation:** 2
**Contribution:** 2
**Rating:** 2
**Confidence:** 4

**Summary:**

The paper introduces dVLA, a diffusion-based vision-language-action model that integrates multimodal chain-of-thought reasoning into robotic control. By unifying visual reasoning, textual understanding, and action prediction within a single diffusion framework, dVLA aims to address gradient conflicts and cross-modal inconsistency present in prior VLA systems. The model demonstrates strong results on the LIBERO benchmark and real-world manipulation tasks, with notable performance gains attributed to multimodal chain-of-thought reasoning and acceleration strategies.

**Strengths:**

1.	The paper proposes a unified diffusion-based objective for jointly optimizing vision, language, and action, which is a conceptually elegant and potentially impactful framework for robotic reasoning.
2.	The authors conduct comprehensive experiments on both simulation (LIBERO) and real-world tasks, showing consistent improvements over several competitive baselines.

**Weaknesses:**

1.	A major weakness is that several recent works already integrate image/video generation and multimodal reasoning for robotic control, yet the paper fails to adequately discuss or compare against these, which undermines claims of novelty. Examples include:
  1.  Zhu et al. Unified World Models: Coupling Video and Action Diffusion for Pretraining on Large Robotic Datasets (RSS 2025)
  2.  Li et al. Unified Video Action Model (RSS 2025)
  3.  Cheang et al. GR-2: A Generative Video-Language-Action Model with Web-Scale Knowledge for Robot Manipulation (Arxiv 2024)
2.	It remains unclear whether the reported performance gains stem from the diffusion-based co-training of image generation or from the proposed multimodal chain-of-thought reasoning. Prior works have already demonstrated benefits from visual co-training, so an ablation or comparative analysis isolating these factors is needed.
3.	The paper states that previous co-training approaches suffer from gradient conflicts but does not explicitly articulate how dVLA reconciles this issue beyond the unified objective. There is insufficient explanation of how knowledge preservation (e.g., for general VQA or world understanding) is maintained after robotics-specific fine-tuning.
5.	While the experiments are extensive, the evaluation scope focuses primarily on short to mid-horizon manipulation tasks. It would strengthen the paper to show that the proposed multimodal CoT scales to complex, compositional, or long-horizon planning scenarios.
6.	The visual CoT examples, while compelling, appear anecdotal. More quantitative evaluation of reasoning correctness or alignment between predicted and executed subgoals would enhance the paper’s rigor.

**Questions:**

1.	The paper could benefit from a clearer theoretical or empirical motivation for why diffusion modeling is particularly advantageous for multimodal alignment compared to autoregressive methods.

---

> ### Author Response · Authors · 2025-12-03
> **Response (1/2)**
>
> ### 1. Clear discussion between dVLA and other models (UWM, UVAM, GR2).
> Thank you for pointing out. The major difference is that dVLA is a vision-language-action model capable of multimodal generation and understanding while UWM, UVAM and GR2 are only world models with action generation ability.
>
> ### 2. Ablation study on only visual image generation.
>
> Thank you for your suggestion. We added an ablation study on the effectiveness of visual image generation as well as two additional tasks: 'Stack Cubes' and 'Place Toy in Drawer'. Note that 'Place Toy in Drawer' is a long-horizon task that requires three steps to complete: (1) pull out the drawer, (2) place the toy inside, and (3) push the drawer back.
>
> As shown in the table below, enabling only visual CoT on top of vanilla dVLA (“Vanilla dVLA (w/ visual CoT)”) already brings clear gains over the baseline without any CoT across all three tasks (e.g., from 55%→70% on Bin Picking and 60%→80% on Stack Cubes), demonstrating that visual subgoal generation alone is beneficial for control. Our full dVLA with both textual and visual MMCoT further improves performance, especially on the long-horizon 'Place Toy in Drawer' task (from 55% to 70%), indicating that textual CoT is particularly helpful for multi-step reasoning.
>
> | Methods / Tasks              | Textual MMCoT | Visual MMCoT | Bin Picking SR  | Stack Cubes SR | Place Toy in Drawer SR |
> |-----------------------------|:-------------:|:------------:|:----------------------:|:--------------:|:-----------------------:|
> | Vanilla dVLA                | ✗ | ✗ | 11/20 (55%) | 12/20 (60%) | 10/20 (50%) |
> | Vanilla dVLA (w/ visual CoT)| ✗ | ✓ | 14/20 (70%) | 16/20 (80%) | 11/20 (55%) |
> | **dVLA (ours)**             | ✓ | ✓ | **15/20 (75%)** | **16/20 (80%)** | **14/20 (70%)** |
>
> ### 3. Knowledge preservation after robotics-specific finetuning.
> Thank you for the thoughtful comment. Our primary goal is to leverage the pretrained multimodal diffusion model to improve downstream robotic control, rather than to preserve generic VQA capabilities. We finetune dVLA with a unified multimodal CoT objective that jointly trains visual subgoals, textual reasoning, and actions. This effectively preserves the pretrained multimodal knowledge relates to robotic control in our downstream robotic tasks.
>
>
> ### 4. Performance on long-horizon tasks.
> This is an excellent suggestion. To improve the robustness of our evaluation, we additionally evaluated all methods on two more tasks with more trials (20 per task). 'Place Toy in Drawer' is a particularly challenging long-horizon task that requires first pulling out the drawer, then placing the toy inside, and finally pushing the drawer back. As shown below, our dVLA achieves a 70% success rate on Place Toy in Drawer, outperforming GR00T by an absolute margin of 30 percentage points, which demonstrates that multi-modal CoT enables dVLA to handle long-horizon tasks effectively.
>
> | Methods / Tasks              | Textual MMCoT | Visual MMCoT | Bin Picking SR | Open Box SR | Hang Cups SR | Pick & Place Object SR | Stack Cubes SR | Place Toy in Drawer SR | Average SR      |
> |-----------------------------|:-------------:|:------------:|:--------------:|:-----------:|:------------:|:----------------------:|:--------------:|:-----------------------:|:---------------:|
> | **Continuous Action Policy**|               |              |                |             |              |                        |                |                         |                 |
> | Diffusion Policy (Chi et al., 2023) | ✗ | ✗ | 5/20  | 7/20  | 10/20 | 6/20  | 11/20 | 7/20  | 46/120 (38.3%) |
> | GR00T (Bjorck et al., 2025)         | ✗ | ✗ | 9/20  | 9/20  | 8/20  | 9/20  | 10/20 | 8/20  | 53/120 (44.2%) |
> | **Discrete Action Policy**  |               |              |                |             |              |                        |                |                         |                 |
> | OpenVLA (Kim et al.)        | ✗ | ✗ | 4/20  | 7/20  | 9/20  | 10/20 | 6/20  | 4/20  | 40/120 (33.3%) |
> | Vanilla dVLA                | ✗ | ✗ | 11/20 | 10/20 | 12/20 | 12/20 | 12/20 | 10/20 | 67/120 (55.8%) |
> | **dVLA (ours)**             | ✓ | ✓ | **15/20** | **11/20** | **13/20** | **16/20** | **16/20** | **14/20** | **85/120 (70.8%)** |

---

> > ### Author Response · Authors · 2025-12-03
> > **Response (2/2)**
> >
> > ### 5. Quantitative evaluation results on alignment between predicted subgoals and executed observations.
> >
> > Thanks for pointing this out. We evaluate the alignment between predicted visual subgoals and executed observations using SSIM and PSNR, two standard metrics in image generation. For each LIBERO task, we run 500 evaluation rollouts, and compute SSIM/PSNR between the predicted subgoal frame and the corresponding future observation at the same timestep, then average the scores within each suite. As shown below, dVLA achieves an average SSIM of 0.69 and PSNR of 20.62 dB across four LIBERO suites. For comparison, in challenging image prediction/generation settings, SSIM in the 0.6–0.7 range and PSNR around 20–25 dB are typically considered reasonable. In short, dVLA achieves reasonably strong alignment between predicted subgoal images and the corresponding executed observations.
> >
> > |                |  SSIM  |  PSNR  |
> > |----------------|--------|--------|
> > | LIBERO-SPATIAL | 0.66   | 20.90  |
> > | LIBERO-OBJECT  | 0.73   | 20.42  |
> > | LIBERO-GOAL    | 0.69   | 21.09  |
> > | LIBERO-LONG    | 0.67   | 20.05  |
> > | Average        | **0.69**   | **20.62**  |
> >
> > ### 6. Why diffusion modeling is particularly advantageous for multimodal alignment compared to autoregressive methods.
> >
> > Thanks for your comment. Diffusion methods facilitate multimodal alignment through bidirectional modeling enabled by masked-token prediction, where all modalities interact mutually during denoising. In contrast, the fixed sequential generation of autoregressive methods often results in a unidirectional mapping from one modality to another, which inherently limits effective cross-modal integration.

---

### Official Review · Reviewer_qBYf · 2025-10-30

**Soundness:** 3
**Presentation:** 3
**Contribution:** 3
**Rating:** 6
**Confidence:** 3

**Summary:**

The authors study VLAs, VLMs which have been adapted to output action data. They argue that typical co-training of VLAs tends to degrade either the general common sense knowledge or the ability to learn robot actions, with it difficult to learn both. Additionally, VLMs are able to do multimodal generation, an underutilized modality. The authors proposed training a VLA to predict subgoal images and subgoal text, given the current image and a high level language instruction. This is trained via masking tokens across all prediction modalities (image, text, action), training reconstruction from one modality to the others. This is called dVLA, since diffusion is used for predicting all modalities

Adding subgoal image and text prediction increases the inference times of dVLA, so there is additional work on KV caching to improve the real world inference speed. Results are tested on a Franka robot.

**Strengths:**

It's been shown in other contexts that adding additional prediction tasks improves the amount of signal extractable from a data source, and so adding additional image and text prediction tasks before action prediction makes sense. The generated visual CoT images are interesting generations as well. The paper is clear on how many trials were done per eval, which is good to see. There are quite a few baselines. The inference improvements are the kind of useful work which is nice to see. Gains are seen in the hardest real-world task setup.

**Weaknesses:**

As noted by the paper, this approach increases the inference time of the method, and although this is mitigated by the prefix attention + KV caching, these improvements could have also been added to the base model which does not do multimodal predictions. So although the paper shows improvement over not using visual CoT, it feels like there is a missing comparison to a larger base VLM with KV caching that does not predict intermediate subgoals.

At its core, the paper is about adding extra subgoal predictions on top of a FAST-based diffusion setup, and this just feels like a minor idea, given the past history of CoT in LLMs and subgoal prediction in robot learning, as well as the only minor improvements seen in LIBERO benchmark.

Some typos (Vallina dVLA --> Vanilla dVLA)

**Questions:**

In Figure 2, are the visual and textual subgoals generated outputs of the model, or are they examples of what subgoals could be? It was not clear to me if "illustrative example" meant "is an actual generation", especially given some of the artifacting in the generated images of Figure 4.

If there are videos of real robot execution, that would also be appreciated.

---

> ### Author Response · Authors · 2025-12-03
> **Response**
>
> ### 1. Inference time comparison with VLM-based VLA model?
>
> Thank you for highlighting the importance of comparing inference speed. In response, we have added two key baselines: (1) a streamlined variant of our dVLA without multimodal chain-of-thought (MMCoT), which uses prefix attention and KV caching, and (2) OpenVLA, a VLM-based model that generates actions autoregressively.As shown below, this efficient dVLA variant maintains strong performance while achieving an inference speed of 5.8 Hz. This rate is substantially faster than that of our full-attention model (1.5 Hz) and the prefix-attention + KV-caching variant with MMCoT (3.0 Hz). It is also faster than the VLM-style baseline, OpenVLA (3.4 Hz). These results demonstrate that our framework can achieve greater inference efficiency than typical VLM-based VLAs.
>
> | Methods                         | MMCoT | Real World Actions / s (↑) |
> |---------------------------------|:-----:|:--------------------------:|
> | Full Attention                  |  ✓    | 1.5 Hz                     |
> | Prefix Attention + KV Caching   |  ✓    | **3.0 Hz**                 |
> | Prefix Attention + KV Caching   |  ✗    | **5.8 Hz**                 |
> | OpenVLA                         |  ✗    | **3.4 Hz**                 |
>
>
> ### 2. At its core, the paper is about adding extra subgoal predictions on top of a FAST-based diffusion setup, and this just feels like a minor idea, given the past history of CoT in LLMs and subgoal prediction in robot learning, as well as the only minor improvements seen in LIBERO benchmark.
>
> Thanks for raising this point. We agree that CoT and subgoal prediction themselves are not new, and our contribution is not simply adding an extra subgoal head on top of FAST. The key idea in dVLA is to build a unified VLA model where visual subgoals, textual reasoning, and actions are generated within the same diffusion-based model and optimized under a single objective. This unified formulation directly avoids the gradient conflicts observed in prior work that combines autoregressive CoT with diffusion/flow-matching policies [1,2], and leads to more coherent vision–language–action behavior, especially on long-horizon and real-robot tasks beyond what is reflected by the aggregate LIBERO scores.
>
> ### 3. What does "illustrative example" means?
>
> Thank you for highlighting this. The illustrative examples in Figure 2 are not actual generated outputs of our model. Instead, they are subgoals selected from the training data to exemplify the expected predictions of dVLA.
>
> [1]. Knowledge Insulating Vision-Language-Action Models: Train Fast, Run Fast, Generalize Better
> [2]. ChatVLA-2: Vision-Language-Action Model with Open-World Embodied Reasoning from Pretrained Knowledge

---

### Official Review · Reviewer_eGfQ · 2025-10-31

**Soundness:** 2
**Presentation:** 3
**Contribution:** 2
**Rating:** 4
**Confidence:** 4

**Summary:**

This paper introduces dVLA, a novel vision-language-action (VLA) framework for robotic control built upon a discrete diffusion language model (DLM). The central contribution is the use of a **unified diffusion-based objective** to jointly optimize visual reasoning, language comprehension, and robotic action generation. The authors frame this as a solution to the gradient conflicts  and architectural gaps  that arise in traditional auto-regressive models, especially when trying to integrate generative capabilities like visual "Chain-of-Thought" (CoT).

The dVLA model discretizes all input and output modalities—including current images, text instructions, robot state, future subgoal images (visual CoT), textual reasoning (textual CoT), and action chunks—into a unified token sequence. It is then trained with a discrete diffusion objective to reconstruct randomly masked tokens from any modality, given the unmasked ones. The authors demonstrate through experiments in both simulation (LIBERO)  and the real world (Franka robot)  that this multimodal CoT approach significantly outperforms existing VLA models. To address the inherent latency of diffusion models, the paper also introduces acceleration strategies, combining a prefix attention mask with KV caching to achieve a ~2x speedup for practical deployment.

**Strengths:**

1.  **Novel and Sound Methodology:** The core idea of using a unified discrete diffusion objective to harmonize perception, reasoning (both visual and textual), and action is elegant and well-motivated. It directly addresses a clear and significant limitation in prior work: the difficulty of integrating generative reasoning (like predicting future images) with action prediction in a single, non-conflicting framework. Building this upon a discrete diffusion model (MMaDA)  and employing discrete tokenization for all modalities (including actions via FAST ) is a technically sound and modern approach.

2.  **Strong Empirical Evaluation:** The paper provides a comprehensive set of experiments that validate the proposed method. dVLA achieves state-of-the-art (SOTA) performance on the LIBERO benchmark, outperforming a wide range of both continuous and discrete action policies. Furthermore, it demonstrates superior performance in a multi-task, real-world setting with four distinct tasks, showing the highest average success rate (65%) compared to all baselines.

3.  **Compelling Ablation Study:** The authors effectively isolate the impact of their primary contribution, the multimodal CoT. The comparison between the full `dVLA` and "Vanilla dVLA" (which lacks the explicit CoT)  shows a substantial performance improvement: a +6.6% average success rate on LIBERO  and a +12.5% improvement in the real-world tasks. This provides strong evidence that the joint generation of visual and textual subgoals is a key driver of the model's success.

4.  **Focus on Practicality (Acceleration):** The paper commendably addresses the primary practical drawback of using diffusion models for robotics: slow inference speed. The authors propose and evaluate two acceleration strategies (Prefix Attention Mask and KV Caching) , demonstrating a ~2x inference speedup with only a marginal drop in task performance. This is a crucial step for making the method viable in real-time settings.

**Weaknesses:**

1.  **Unclear Explanation of Acceleration Method:** The description of the "Prefix Attention Mask" in Section 3.4 is confusing and difficult to follow. The text describes a "blockwise causal attention mask with two blocks"  and "full bidirectional attention within each block" , but then states "tokens in one block restricted from attending to tokens in subsequent blocks". This description is ambiguous. For the KV caching of the prefix (Block 1) to be useful, the generated tokens (Block 2) must be able to *attend to* the prefix tokens. The diagram in Figure 1 appears to show a standard prefix-LM, but the text description does not clearly align with this. This methodological detail is critical and needs to be clarified.

2.  **Marginal SOTA and Baseline Confusion on LIBERO:** While dVLA achieves SOTA on LIBERO (96.4%) , the margin over the "Discrete Diffusion VLA" baseline (96.3%)  is only 0.1%. This is a razor-thin difference and suggests that the core benefit might come from using *any* discrete diffusion VLA, rather than the paper's specific multimodal CoT contribution. This is confounded by the "Vanilla dVLA" baseline (89.8% SR). It is not clear what the difference is between "Vanilla dVLA"  and "Discrete Diffusion VLA". Both appear to be discrete diffusion VLAs without the multimodal CoT, yet their performance is drastically different. The paper needs to explicitly define these baselines and directly address why the multimodal CoT provides only a 0.1% boost over a seemingly similar SOTA diffusion model, even though it provides a 6.6% boost over the "Vanilla dVLA" ablation.

3.  **Contradictory Information on Textual CoT Usage:** There appears to be a contradiction regarding the use of textual CoT. In Section 3.5, the paper states, "For simpler tasks, we omit language reasoning to further speed up inference". However, Table 1 (LIBERO results) clearly shows `dVLA` using *both* textual and visual CoT (it has a "✓" in both columns). This implies the LIBERO tasks were *not* considered "simpler tasks." This should be clarified: was textual CoT used for all LIBERO tasks? If so, how was this reasoning data generated, given that the SEED-1.5VL method was described as "designed for long-horizon tasks such as bin picking"?

4.  **Limited Scale of Real-World Evaluation:** While the inclusion of real-world experiments is a significant strength, the evaluation scale is limited (10 trials per task). With these few trials, the 65% success rate (26/40) for `dVLA`  is promising but may not be statistically robust. For example, the 7/10 success on "Bin Picking" and "Hang Cups"  is good, but more trials would be needed to build stronger confidence in the method's reliability compared to baselines.

**Questions:**

1.  **Prefix Mask Clarification:** Could you please provide a clearer explanation of the "Prefix Attention Mask" (Sec 3.4)? Does Block 2 ($[o_{\mathfrak{g}oal},r,a_{chunk}]$)  attend to Block 1 ($[o,l, s]$)? If so, how should we interpret the text "tokens in one block restricted from attending to tokens in subsequent blocks"?

2.  **Baseline Comparison (Vanilla dVLA vs. Discrete Diffusion VLA):** What is the exact architectural and training objective difference between your "Vanilla dVLA" baseline (89.8% LIBERO SR)  and the "Discrete Diffusion VLA" (Liang et al., 2025) baseline (96.3% LIBERO SR)? Why does the latter perform so much better, and why is the margin between your full `dVLA` (96.4%)  and "Discrete Diffusion VLA" only 0.1%?

3.  **Textual CoT Usage on LIBERO:** Can you confirm if textual CoT was used for the LIBERO benchmark results in Table 1? If so, how does this align with the statement that language reasoning is omitted for "simpler tasks", and how was this textual reasoning data generated for the LIBERO dataset?

4.  **Inference Procedure:** During inference, are the visual CoT, textual CoT, and action tokens all generated *simultaneously* in one parallel denoising process (as suggested by Fig 1 and the unified objective)? Or is the process sequential (as implied by Sec 3.3, "generates two parallel outputs... Subsequently, dVLA grounds these... to produce a concrete and executable action" )?

I am open to raising the score if above concerns could be well explained and addressed.

---

> ### Author Response · Authors · 2025-12-03
> **Response**
>
> ### 1. Clear description on attention mask strategy.
> Thanks for pointing this out. To properly separate the input context from the tokens being generated, we partition the token sequence into two blocks. The first block contains the input visual and textual tokens, while the second block contains all generated visual, textual, and action tokens. We apply a prefix attention mask to enforce that tokens in the first block cannot attend to those in the second block. Conversely, tokens in the second block can attend to all tokens in the first block. Importantly, full bidirectional attention is always maintained within each block itself.
>
> ### 2. What is the difference between Vanilla dVLA and Discrete Diffusion VLA.
> Thank you for highlighting this. First, we would like to clarify that Discrete Diffusion VLA and dVLA are concurrent works. What distinguishes dVLA is its motivation to create a unified vision-language-action model by optimizing visual reasoning, language comprehension, and robotic actions simultaneously through a single diffusion-based objective. Previous work[1,2] has shown that using autoregressive methods for CoT generation and diffusion methods (or flow matching) for action prediction can lead to gradient conflicts. To overcome this challenge, we tokenize the vision, language, and action modalities into a shared discrete space, constructing a multimodal CoT that better captures the underlying vision-action alignment.
>
>
> ### 3. Was textual CoT used for all LIBERO tasks and how to generate such reasoning data?
> Yes, textual CoT is used for all LIBERO tasks in our main results. We generate this reasoning using SEED-VL 1.5: given the current frame and instruction, we prompt the model to produce a short, step- by-step description of the next subgoal (e.g., "the cup is on the left, so the arm should move left to grasp it"). For simple, short-horizon tasks, we observe smaller but still consistent gains from textual CoT, whereas for long-horizon tasks the improvements are substantial. Therefore, in latency-sensitive settings, omitting textual CoT for simple tasks is a practical knob to trade off a small performance drop for higher inference speed.
>
> ### 4. Limited scale of real-world evaluation.
> Thank you for the excellent suggestion. We evaluated each task 10 trials per method which is a common practice in robotics[3,4]. To further improve the robustness of our evaluation, we have additionally tested all methods on two new tasks with 20 trials each. Among these, 'Place Toy in Drawer' is a particularly challenging, long-horizon task: it requires the agent to first pull open the drawer, then place the toy inside, and finally push the drawer closed. As shown in the results, our dVLA consistently outperforms all baseline methods across every task. It achieves an average success rate of 70.8%, representing an absolute improvement of 26.6 percentage points over the second-best baseline (GR00T).
>
> | Methods / Tasks              | Textual MMCoT | Visual MMCoT | Bin Picking SR | Open Box SR | Hang Cups SR | Pick & Place Object SR | Stack Cubes SR | Place Toy in Drawer SR | Average SR      |
> |-----------------------------|:-------------:|:------------:|:--------------:|:-----------:|:------------:|:----------------------:|:--------------:|:-----------------------:|:---------------:|
> | **Continuous Action Policy**|               |              |                |             |              |                        |                |                         |                 |
> | Diffusion Policy (Chi et al., 2023) | ✗ | ✗ | 5/20  | 7/20  | 10/20 | 6/20  | 11/20 | 7/20  | 46/120 (38.3%) |
> | GR00T (Bjorck et al., 2025)         | ✗ | ✗ | 9/20  | 9/20  | 8/20  | 9/20  | 10/20 | 8/20  | 53/120 (44.2%) |
> | **Discrete Action Policy**  |               |              |                |             |              |                        |                |                         |                 |
> | OpenVLA (Kim et al.)        | ✗ | ✗ | 4/20  | 7/20  | 9/20  | 10/20 | 6/20  | 4/20  | 40/120 (33.3%) |
> | Vanilla dVLA                | ✗ | ✗ | 11/20 | 10/20 | 12/20 | 12/20 | 12/20 | 10/20 | 67/120 (55.8%) |
> | **dVLA (ours)**             | ✓ | ✓ | **15/20** | **11/20** | **13/20** | **16/20** | **16/20** | **14/20** | **85/120 (70.8%)** |
>
> [1]. Knowledge Insulating Vision-Language-Action Models: Train Fast, Run Fast, Generalize Better
> [2]. ChatVLA-2: Vision-Language-Action Model with Open-World Embodied Reasoning from Pretrained Knowledge
> [3]. OpenVLA: An Open-Source Vision-Language-Action Model
> [4]. π0: A Vision-Language-Action Flow Model for General Robot Control
>
> ### 5. Are the visual CoT, textual CoT, and action tokens all generated simultaneously?
> Thanks for pointing this out. They are not generated simultaneously. DVLA first generates visual CoT, then generates textual CoT and aciton tokens simutaneously.

---

### Meta-Review · Area_Chair_6DGM · 2026-01-06

**Summary:**

Initial reviews were low to mixed (2/4/4/6). While reviewers praised the unified objective and acceleration strategies, they raised significant concerns regarding baselines, ablations, and evaluations. I encourage the authors to resubmit the work after a major overhaul as the proposed method appears promising.

**Reviewer Concerns:**

See summary above & reviewer score notes below.

**Reviewer Scores:**

Reviewer eGfQ: 4 --> perhaps 5. concerns have been addressed, and the reviewer explicitly mentioned that they are open to increasing the score.
Reviewer qBYf: 6 changes unlikely
Reviewer pWAW: 4 --> perhaps 5 main weaknesses / questions have been addressed.
Reviewer eZGi: 2 --> trending positive, but likely still reject. The authors addressed the main grounds for rejection by adding long-horizon tasks and further discussions and results.

---

### Decision · Program_Chairs · 2026-01-26

Reject